# BANDITS WITH ANYTIME KNAPSACKS

## ABSTRACT

We consider bandits with anytime knapsacks (BwAK), a novel version of the BwK problem where there is an *anytime* cost constraint instead of a total cost budget. This problem setting introduces additional complexities as it mandates adherence to the constraint throughout the decision-making process. We propose SUAK, an algorithm that utilizes upper confidence bounds to identify the optimal mixture of arms while maintaining a balance between exploration and exploitation. SUAK is an adaptive algorithm that strategically utilizes the available budget in each round in the decision-making process and skips a round when it is possible to violate the anytime cost constraint. In particular, SUAK slightly under-utilizes the available cost budget to reduce the need for skipping rounds. We show that SUAK attains the same problem-dependent regret upper bound of $O(K \log T)$ established in prior work under the simpler BwK framework. Finally, we provide simulations to verify the utility of SUAK in practical settings.

## 1 INTRODUCTION

Multi-armed bandits (MAB) is one of the fundamental problems in the field of sequential decision-making under uncertainty. In its essence, it is a problem setting where an agent must strategically allocate resources among the arms to maximize cumulative reward over time, navigating the trade-off between gathering information about uncertain arms (exploration) and exploiting known information to optimize immediate rewards (exploitation). This problem finds applications across diverse domains, including reinforcement learning (Intayoad et al., 2020), online advertising (Slivkins, 2013), clinical trials (Villar et al., 2015), and resource allocation (Soare, 2015).

The *Bandits with Knapsacks* (BwK) problem, introduced by Badanidiyuru et al. (2013), is an extension of the classical multi-armed bandit problem, with the additional constraint of limited resource capacity akin to the *knapsack* problem (Tran-Thanh et al., 2012). In this scenario, an agent is confronted with a set of arms, each associated with an *unknown* reward and cost distribution. Unlike the traditional bandit setting, selecting an arm incurs both a reward and a cost here, and the agent's objective is to maximize the total reward while respecting the total capacity constraint of the knapsack. The BwK problem encapsulates the trade-off between exploration and exploitation while managing resource constraints, presenting a rich framework with applications such as online advertising (Avadhanula et al., 2021; Badanidiyuru et al., 2018), dynamic resource allocation (Kumar & Kleinberg, 2022), and personalized recommendation engines (Yu et al., 2016).

In this paper, we consider a specific variant of this problem, which we name as the bandits with anytime knapsacks (BwAK) problem; where instead of a total cost budget, there is an anytime constraint on the average cost. This problem setting introduces an additional level of complexity as a mixture strategy needs to be employed to be able to pull arms with mean costs higher than the average cost budget without violating the anytime constraint. The main goal of our work is to develop new algorithms for this framework that achieve as much *cumulative* reward as possible.

### 1.1 APPLICATIONS

The formulation of the anytime constraint considered here has broad applications across various fields. A notable example is inventory management, where a factory produces goods at a constant rate and seeks to maximize revenue by selling to buyers in a marketplace, where bids consisting of price and order size are placed. Our anytime constraint is especially relevant in such scenarios since having a negative inventory is not possible. An important aspect of this constraint is that it introduces

the trade-off between exploiting the available inventory, and skipping a round to accumulate more inventory in an effort to capture bids with higher order sizes. This highlights the added complexity of our problem setting and underscores its broader applicability across a range of settings beyond the standard BwK framework.

Another example is online advertising, where an advertiser sets a daily budget limit to prevent over-spending. In this context, the 'arms' represent different ad campaigns or strategies, each with varying costs that can be selected for the day. The reward can be modeled as the daily revenue generated from clicks or the number of users who subscribe. Further, in portfolio management, our anytime constraint can represent the maximum amount that the customer is willing to invest in a month, and arms can model different investment options. One last example is in satellite systems, where solar panels generate energy and excess energy can be stored in a battery. Here, $c$ can correspond to the energy generated per unit time, and arms can correspond to different tasks that need to be performed, with their rewards reflecting the importance or outcome of the tasks. The costs associated with each arm can than represent the energy consumed to complete the task.

## 1.2 CONTRIBUTIONS

1. **Formulation:** To our knowledge, this work is the first to consider a multi-armed bandit with knapsacks (BwK) setting with an *anytime* cost constraint.

2. **SUAK Algorithm:** SUAK utilizes the upper confidence bounds to explore the best base that solves the problem, and also strategically under-utilizes the available budget to limit the number of rounds that are skipped when satisfying the anytime cost constraint.

3. **Regret Upper Bound for SUAK:** We provide upper bounds on the expected cumulative regret of SUAK for this problem setting, and establish that it scales as $O(K \log T)$.

Related works is provided in §.

## 2 PROBLEM STATEMENT

### 2.1 THE BANDITS WITH ANYTIME KNAPSACKS (BWAK) MODEL

We consider a $K$-armed stochastic bandit problem with the set of base arms $[K]$, where pulling arm $i \in [K]$ in round $t$ is associated with a random cost, $\rho_i(t)$; drawn from a probability distribution supported in $[0, 1]$ with mean $\rho_i$, that is independent of the costs of other arms. After pulling arm $i$ in round $t$, the agent receives a random reward, $r_i(t)$; drawn from a probability distribution supported in $[0, 1]$ with mean $\mu_i$, that is independent of the rewards of other arms. At each round $t$, the agent has the option of skipping by not pulling any of the $K$ arms. We model this decision by introducing an arm which has a cost and reward of 0, as arm $K + 1$, which is known as the *null arm* in BwK literature. We let $\boldsymbol{\rho} = [\rho_1, \cdots, \rho_K, 0]^T$ and $\boldsymbol{\mu} = [\mu_1, \cdots, \mu_K, 0]^T$ denote the mean cost vector and the mean reward vector of the arm set $[K + 1]$, respectively. Throughout this paper, we use bold symbols to denote vectors or matrices. $\Delta_{K+1}$ is used to denote the $K + 1$-dimensional probability simplex. We let $i^* := \arg\max_{i \in [K]} \mu_i/\rho_i$ denote the arm with highest mean reward per cost, and let $i^{**} := \arg\max_{i \in [K]} \mu_i$ denote the arm with the highest mean reward. For simplicity, we assume that there is only one arm with highest mean reward per cost and also there is only one arm with highest mean reward.

Let $i(t)$ be the arm pulled by the agent in round $t$, $r(t)$ represent the reward received in round $t$, and $c(t)$ represent the cost incurred in round $t$. Also let $N_i(t)$ denote the total number of times arm $i$ has been pulled up to round $t$. Further, define $S_c(t) = \sum_{s=1}^{t} c(s)$ and $\bar{c}(t) = S_c(t)/t$ as the cumulative cost and the average cost incurred until round $t$. Let $\bar{\rho}_i(t) = \sum_{s=1}^{t} \rho_i(s) \cdot \mathbb{1}\{i(t) = i\}/N_i(t)$ be the empirical average cost of arm $i$ at round $t$, and similarly let $\bar{\mu}_i(t) = \sum_{s=1}^{t} \mathbb{1}\{i(t) = i\} \cdot \mu_i(s)/N_i(t)$ be the empirical average reward. We assume that there is an average cost budget of $c$ per round that cannot be exceeded at any round, which we refer to as the anytime cost constraint. The agent aims to maximize cumulative reward received under this constraint. This can formally be expressed as:

$$\text{maximize } F(t) = \mathbb{E}\left[\frac{1}{t}\sum_{s=1}^{t} r(s)\right] \quad \text{s.t. } \frac{\sum_{s=1}^{u} c(s)}{u} \leq c \, , \forall \, u \leq t.$$

This setting represents many practical applications as discussed in §1.1.

**Linear Relaxation.** Following the prior work, we consider the following linear relaxation:

$$OPT_{LP}(T) = \max_{\boldsymbol{\pi}} T \cdot \boldsymbol{\mu}^T \boldsymbol{\pi} \tag{1}$$

$$\text{s.t. } \boldsymbol{\rho}^T \boldsymbol{\pi} < c,$$

$$\pi_i \geq 0, \forall i \in [K+1].$$

where the vector $\boldsymbol{\pi}$ represents the policy which defines the fraction of time an arm will be pulled. In any policy, there will be at most two arms that have nonzero $\pi_i$ values since there are two constraints in the problem. We refer to a set consisting of at most two arms as a base. We denote the set of all possible valid bases (where valid means that the average cost less than or equal to $c$ can be reached through a mixture of arms in the base) as $\mathbb{V}$. Note that for simplicity, we assume that the arms in a base are ordered so that the higher cost arm appears first. We let $\mathcal{I}^i := \{\mathcal{I} \in \mathbb{V} : i \in \mathcal{I}\}$ denote the set of valid bases that include the arm $i$. We let $\boldsymbol{\pi}^*$ denote the optimal solution to (1), and let $r^* := \boldsymbol{\mu}^T \boldsymbol{\pi}^*$ be the optimal reward per round. We also define $\mathcal{I}^* := \{i : \pi_i^* > 0\}$ as the optimal base. The optimal solution of this problem can be divided into three cases. First, if the arm with the highest mean reward has cost less than $c$, i.e. $\rho_{i^{**}} \leq c$; then the optimal base consists of only this arm; hence $\mathcal{I}^* = \{i^{**}\}$, and $\pi_{i^{**}}^* = 1$. In the second case, if $\rho_{i^{**}} > c$, $\rho_{i^*} > c$, then $\mathcal{I}^* = \{i^*, K+1\}$, and the optimal solution is $\pi_{i^*}^* = c/\rho_{i^*}$, and $\pi_{K+1}^* = 1 - \pi_{i^*}^*$. In third case, if $\rho_{i^{**}} > c$, $\rho_{i^*} < c$, then optimal base includes two arms which might or might not include $i^*$ or $i^{**}$.

Let $OPT$ denote the total expected reward of a dynamic policy in $T$ rounds that conforms to a total budget constraint in a total of $T$ rounds as in the standard BwK literature instead of the anytime budget constraint we consider here. It was shown that $OPT_{LP} \geq OPT$ (Badanidiyuru et al., 2013).

Let $REF$ be denote the total expected reward of a dynamic policy in $T$ rounds that conforms to the average cost constraint. This constraint is stricter than the total budget constraint. This can easily be seen as satisfying the anytime constraint in the last round $T$ with $c = B/T$ produces the total budget constraint of $B$ in $T$ rounds. Hence, it holds that $OPT_{LP} \geq OPT \geq REF$. While regret could be defined as the difference expected cumulative reward of SUAK and $REF$, we choose a stronger regret definition by defining it with respect to $OPT_{LP}$ as $R_T = OPT_{LP} - \mathbb{E}[F(T)] = Tr^* - \mathbb{E}[F(T)]$ so that our results can be compared with prior work on the total budget setting.

# 3 THE SUAK ALGORITHM

## 3.1 THE NAIVE APPROACH

Before presenting the SUAK Algorithm, to demonstrate the additional complexities of our problem formulation over the standard BwK setting, and also to serve as a baseline, we present a naive approach which makes it possible to convert any BwK algorithm to our BwAK setting. In this trivial approach, in a given round $t$, we first simply check if it is possible to violate the anytime constraint, and skip the round if it is the case. Otherwise, we let the BwK algorithm pull an arm. To demonstrate this more concretely, we use the *One Phase* Algorithm in Li et al. (2021), and add skipping behaviour such that a round is $t$ skipped if $S_c(t-1) + 1 > c \cdot t$. The implementation with this skipping rule, which we call as the *One Phase Skip (OPS)* Algorithm, is given in Algorithm 1.

In this algorithm, the initialization phase consists of sampling each arm once while using skips to prevent violation of the constraint. After this phase, we utilize a skipping mechanism in lines 5 - 6, and if the round is not skipped, the algorithm proceeds to solving the linear programming problem in line 8. In this LP, $\boldsymbol{\mu}^U(t)$ is the UCB of arm reward at round $t$, $\boldsymbol{\rho}^L(t)$ is the LCB of arm cost at round $t$, and $B_r(t) = cT - S_c(t-1)$ is the total remaining budget in round $t$. Since UCB values are used, the solution of LP gives the optimistically best policy according to the UCB principle. This policy is normalized to a probability distribution, and the arm is selected using this probability.

To show that this naive approach might suffer a large regret due to large number of skips, we run simulations on the following problem instance with $K+1 = 4$ arms where $\mu = [0.45, 0.7, 0.8, 0]$; and $\rho = [0.25, 0.75, 0.8, 0]$. Except for the null arm, the arm reward and cost values are independently sampled from a Beta distribution with parameters $\alpha = \mu * 10, \beta = (1-\mu) * 10$. The average cost budget per round is $c = 0.5$. For SUAK, we take $\omega = 0.143$. We average results from 20 sim-

---

**Algorithm 1** The Naive Approach: One Phase Skip Algorithm

---

1: **Input:** Average cost target $c$, number of rounds $T$

2: **Initialize:** Sample each arm once while skipping accordingly so that $\forall t \leq t_{\text{init}}$, $S_c(t-1) + 1 \leq c \cdot t$

3: Set $t = 1$

4: **for** each round $t > t_{\text{init}}$ **do**

5:     **if** $S_c(t-1) + 1 > c \cdot t$ **then**

6:         Skip the round

7:     **else**

8:         Solve the following LP:
$$\tilde{\boldsymbol{\pi}} = \arg\max_{\boldsymbol{\pi}} \quad \langle \boldsymbol{\mu}^U(t-1), \boldsymbol{\pi} \rangle$$
$$\text{s.t.} \quad \langle \boldsymbol{\rho}^L(t-1), \boldsymbol{\pi} \rangle \leq B_r(t)$$
$$\boldsymbol{\pi} \geq \boldsymbol{0}$$

9:         Normalize $\tilde{\boldsymbol{\pi}}$ into a probability and randomly play an arm from this probability

10:     **end if**

11:     Update $\boldsymbol{\rho}^L(t), \boldsymbol{\mu}^U(t)$, and $\boldsymbol{B}^{(t)}$

12:     Update $t = t + 1$

13: **end for**

---

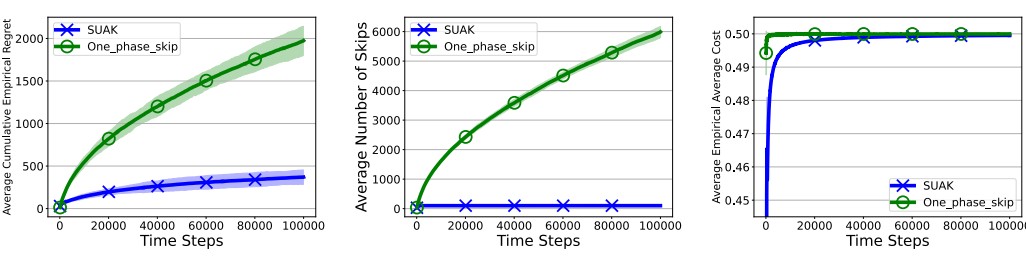

Figure 1: The plots of cumulative empirical regret (Left), number of skips (Middle), and cumulative number of skips (Right); averaged over 20 different trials.

ulation runs, each with $100,000$ rounds. We compare the results with results from SUAK which we propose in §3.2. The simulation results are given in Figure 1. The shaded areas in the plots represent error bars with one standard deviation. It can be seen that the number of skips of One Phase Skip algorithm exhibits sublinear growth and it is much higher than the number of skips of SUAK, which results in higher regret compared to SUAK. Hence, this demonstrates that merely adding skips to a BwK algorithm and treating it as a BwAK algorithm is not a viable solution. In view of this, in the next section, we present SUAK, an algorithm that strategically under-utilizes the available budget to reduce the number of skips needed, and achieve smaller regret.

### 3.2 THE SUAK ALGORITHM

We propose an algorithm called *Strategic Under-utilization for Anytime Knapsacks* (SUAK) that utilizes upper and lower confidence bounds of the arm rewards and costs using the UCB principle to upper bound the reward that can be obtained from a particular base. SUAK also uses skipping a round to satisfy the anytime cost constraint, and targets an average cost of $c - \log t/(\omega^2 t)$ to limit the number of skips, where $\omega$ is defined in Assumption 2. The pseudo-code is provided in Algorithm 2.

The algorithm works as follows. First, SUAK is initialized by sampling each arm once. To prevent SUAK from exceeding the targeted average of $c - \log t/(\omega^2 t)$ during initialization, skipping is employed in rounds $t \leq t_{\text{init}}$ whenever $S_c(t-1) + 1 > c \cdot t - \log(t)/\omega^2$. We define $t_{\text{init}}$ as the round where initialization ends (when a sample is obtained from each arm). After this initialization step, the algorithm works as follows. In every round, first the anytime budget constraint is checked. The round is skipped (null arm is pulled) and the algorithm proceeds to the next round if $S_c(t-1) + 1 > c \cdot t$, i.e. if pulling an arm at that round can violate the constraint. Secondly, if there is uncertainty on whether the mean cost of an arm is less than or greater than $c$, i.e. $\exists l : \varrho_l^L(t) \leq c \leq \varrho_l^U(t)$ where

$$\varrho_l^L(t) := \bar{\rho}_l(t) - 7\sqrt{1.5 \log t/N_l(t)}, \text{ and } \varrho_l^U(t) := \bar{\rho}_l(t) + 7\sqrt{1.5 \log t/N_l(t)};$$

then that arm is pulled and SUAK proceeds to the next round. This step is needed for the anytime cost constraint; it ensures whether the mean cost of arm is above or below $c$ is correctly known, which in turn ensures that the base that SUAK selects for that round includes an arm with cost less than $c$. This step is also needed to establish tighter bounds on the number of times a suboptimal base is selected in the theoretical analysis. To prevent this step from using more than $c$ cost budget per round on average, we define $S_p(t)$ as the sum of all the cost incurred from this step until round $t+1$, and define $N_p(t)$ as the total number of arm pulls due to this step until round $t+1$. The round

---

**Algorithm 2** SUAK: Strategic Under-utilization for Anytime Knapsacks

---

1: **Input:** Average cost target $c$
2: **Initialize:** Sample each arm once while skipping accordingly so that $\forall t \leq t_{\text{init}}$, $S_c(t-1) + 1 \leq c \cdot t - \log(t)/\omega^2$
3: **for** each round $t > t_{\text{init}}$ **do**
4:     **if** $S_c(t-1) + 1 > c \cdot t$ **then**
5:         Skip round $t$
6:         **continue**
7:     **end if**
8:     **if** $S_p(t-1) + 1 > c \cdot N_p(t-1)$ **then**
9:         Skip round $t$
10:         **continue**
11:     **end if**
12:     **if** $\exists l : \varrho_l^L(t) \leq c \leq \varrho_l^U(t)$ **then**
13:         Pull arm $i(t) = l$, observe $r_{i(t)}(t)$, $\rho_{i(t)}(t)$
14:         $S_p(t) = S_p(t-1) + \rho_{i(t)}(t)$
15:         $N_p(t) = N_p(t-1) + 1$
16:         **continue**
17:     **end if**
18:     $\mathcal{S}_t = \{\boldsymbol{\pi} : \boldsymbol{\pi} \in \Delta_{K+1}, \langle \boldsymbol{\pi}, \boldsymbol{\rho}^L(t-1) \rangle \leq c\}$
19:     $\boldsymbol{\pi}(t) = \arg\max_{\boldsymbol{\pi} \in \mathcal{S}_t} \langle \boldsymbol{\mu}^U(t-1), \boldsymbol{\pi} \rangle$
20:     $\mathcal{I}_t = \{i : \pi_i(t) > 0\}$
21:     **if** $|\mathcal{I}_t| = 1$, i.e. $\mathcal{I}_t = \{j(t)\}$ **then**
22:         Pull arm $j(t)$
23:         **continue**
24:     **end if**
25:     $j(t), k(t) = j(t), k(t) \in \mathcal{I}_t : \bar{\rho}_{j(t)}(t) > \bar{\rho}_{k(t)}(t)$
26:     $b(t) = c \cdot t - S_c(t-1) - \log t/\omega^2$
27:     **if** $b(t) > \bar{\rho}_j(t)$ **then**
28:         $p(t) = 1 - \omega$
29:     **else if** $b(t) < \bar{\rho}_k(t)$ **then**
30:         $p(t) = \omega$
31:     **else**
32:         $p_1(t) = \max\left(\frac{b(t) - \bar{\rho}_{k(t)}(t)}{\bar{\rho}_{j(t)}(t) - \bar{\rho}_{k(t)}(t)}, \omega\right)$
33:         $p(t) = \min(p_1(t), 1 - \omega)$
34:         $i(t) = \begin{cases} j(t) \text{ with probability } p(t), \\ k(t) \text{ otherwise} \end{cases}$
35:     **end if**
36:     Pull arm $i(t)$, observe $r_{i(t)}(t)$, $\rho_{i(t)}(t)$
37:     Update $\boldsymbol{\rho}^L(t)$ and $\boldsymbol{\mu}^U(t)$
38: **end for**

---

$t$ is skipped if $S_p(t-1) + 1 > c \cdot N_p(t-1)$. The main objective of this skipping mechanism is to decouple the skips needed to satisfy the anytime cost constraint due to regular arm pulls and the skips needed to satisfy the constraint from this step for ease of theoretical analysis; in practice this skipping mechanism can be ignored. Since the expected number of arm pulls from this step is upper bounded by $O(\log T)$, the skips due to this step will also be $O(\log T)$. After this step, the constraint set is constructed as $\mathcal{S}_t = \{\boldsymbol{\pi} : \boldsymbol{\pi} \in \Delta_{K+1}, \langle \boldsymbol{\pi}, \boldsymbol{\rho}^L(t-1) \rangle \leq c\}$; where

$$\mu_i^L(t) := \text{proj}_{[0,1]}\left(\bar{\mu}_i(t) - \epsilon_i(t)\right), \qquad \mu_i^U(t) := \text{proj}_{[0,1]}\left(\bar{\mu}_i(t) + \epsilon_i(t)\right),$$

$$\rho_i^L(t) := \text{proj}_{[0,1]}\left(\bar{\rho}_i(t) - \epsilon_i(t)\right), \qquad \rho_i^U(t) := \text{proj}_{[0,1]}\left(\bar{\rho}_i(t) + \epsilon_i(t)\right),$$

are the UCB and LCB values of arm costs and rewards; and $\epsilon_i(t) = \sqrt{3 \log T / N_i(t)}$ is the confidence interval. Hence, the constraint set $\mathcal{S}_t$ includes all policies that have an average cost less than $c$ using the optimistic estimates of arm costs (LCB values of arm costs). The empirically best policy at round $t$ is then found using a linear program (LP) as $\boldsymbol{\pi}(t) = \arg\max_{\boldsymbol{\pi} \in \mathcal{S}_t} \langle \boldsymbol{\mu}^U(t-1), \boldsymbol{\pi} \rangle$. Note that using the UCB of the empirical arm reward along with the LCB of empirical arm costs in $\mathcal{S}_t$ produces an upper confidence bound on the reward of a base. The arms that have nonzero $\pi_i(t)$ values are selected as the empirically optimal base arm set for that round, denoted as $\mathcal{I}_t$.

If $\mathcal{I}_t$ consists of a single arm, that arm is pulled. Otherwise, $\mathcal{I}_t = \{j(t), k(t)\}$ will consist of two arms; we denote them as $j(t)$, and $k(t)$; where *wlog* we assume $j(t)$ is the arm with mean cost above $c$. The available budget at that round with respect to the targeted average cost is found as $b(t) = c \cdot t - S_c(t-1) - \log t/\omega^2$. If the available budget $b(t)$ is greater than $\bar{\rho}_{j(t)}(t)$, arm $j(t)$ is pulled with probability $1 - w$, and arm $k(t)$ is pulled otherwise. If $b(t)$ is less than $\bar{\rho}_{k(t)}(t)$, arm $j(t)$ is pulled with probability $w$, and arm $k(t)$ is pulled otherwise. If $\bar{\rho}_{k(t)}(t) \leq b(t) \leq \bar{\rho}_{j(t)}(t)$, arm $j(t)$ is pulled with probability $p(t) = \frac{b(t) - \bar{\rho}_{k(t)}(t)}{\bar{\rho}_{j(t)}(t) - \bar{\rho}_{k(t)}(t)}$ clipped at $w$ from below and $1 - w$ from above; and arm $k(t)$ is pulled otherwise. With this design, each arm in a base is pulled with at least $w$ probability to help explore all arms in a base.

Note that this algorithm is non-stationary as it is adaptive to the available budget at that round. This design is essential as it was shown in Flajolet & Jaillet (2015, Lemma 2) that a non-adaptive design suffers regret of order $\Omega(\sqrt{T})$ even if the optimal solution $\boldsymbol{\pi}^*$ is known unless all arms consume the same deterministic amount of resources at every round. The main intuition behind this result is that the fluctuation of the available budget around its mean at a round $t$ can be as high as $\Omega(1/\sqrt{t})$.

## 3.3 ANALYSIS OF SUAK

We now characterize the performance of the SUAK by providing the theoretical upper bound on the expected cumulative regret. We first provide definitions of arm gaps and state a set of mild assumptions that are required for the theoretical analysis. We refer the readers to the Appendix for detailed proofs of the results presented in this section.

**Definition 3.1.** *The reward gap of an arm is defined as $\Delta_i = \mu_{i^{**}} - \mu_i$.*

**Definition 3.2.** *The gap of a base $\mathcal{I}$ is defined as $\Delta_{\mathcal{I}} = r^* - r_{\mathcal{I}}$, where $r_{\mathcal{I}}$ is the reward value of the solution of (1) when only arms in $\mathcal{I}$ are allowed. We also define $\Delta_{\min,i} := \min_{\mathcal{I} \in \mathcal{I}^i \setminus \mathcal{I}^*} \Delta_{\mathcal{I}}$ as the minimum reward gap of bases that include the arm $i$.*

**Definition 3.3.** *The cost gap of an arm is defined as $\delta_i = |\rho_i - c|$.*

**Assumption 1.** *We define $\delta_{\min} = \min_{i \in [K]} \delta_i$ as the minimum cost gap, and assume $\delta_{\min} > 0$.*

Note that regret depends on the cost gap $\delta_i$ since the algorithm needs to be able to correctly identify if the true mean cost of an arm is above or below $c$. This is in turn needed for the adaptive design since if the empirical average cost is above the targeted cost and if an arm with cost more than $c$ is identified as an arm with cost less than $c$, pulling that arm might lead to over-consuming the targeted budget. Since regret depends on $\delta_{\min}$, $\delta_{\min} > 0$ is needed so that the regret bound is not unbounded.

**Assumption 2.** *We assume that we are given an $\omega > 0$ such that $\omega \leq \delta_{\min}/(2 + \delta_{\min} - c)$.*

Note that for any $\delta_{\min}$ or $c$ value, $\delta_{\min}/(2 + \delta_{\min} - c) \geq \delta_{\min}/3$. Assumption 2 is necessary for the adaptive design of the algorithm in meeting the anytime cost constraint, as we use a cost budget under-utilization of $\log t/\omega^2$ at round $t$ in SUAK to be able to achieve theoretical guarantees. Also, in SUAK, we set the minimum fraction of time an arm in a base will be pulled to $\omega$. With this use, $\omega$ can be understood as the minimum triggering probability ($p^*$), in the probabilistic triggering literature discussed in §3.4. Since the fraction of pulls of a particular arm in a given base can be as low as $\omega$, our regret bounds depend on $\omega$ as in the worst case a base needs to be selected $1/\omega$ times in expectation to acquire one sample of each arm in the base.

Under these assumptions stated above, we obtain the following upper bound on expected regret.

**Theorem 3.1** (Upper Bound on Expected Regret). *Under Assumption 1 and 2; when SUAK is run with a given average cost budget $0 < c \leq 1$, its cumulative expected regret is upper bounded as*

$$R_T \leq \sum_{i=1}^{K} \frac{96 r^* (\frac{\delta_i + 1}{\delta_i})^2 \log T}{\omega \Delta_{\min,i}^2} + \frac{202 K r^* \log T}{c \omega^2} + \frac{3 \pi^2 r^*}{\delta_{min}^2} + R_K + r^* t_{in} = O(K \log T) + O(1) \quad (2)$$

*where $R_K = 5 \pi^2 K^2 / 3$, and $t_{in} = -W\left(-\omega^2 c e^{-\omega^2 K}\right)/(\omega^2 c) = O(1)$ is the upper bound on the number of rounds needed for the initialization phase of SUAK; and $W(\cdot)$ is the Lambert function. Also recall that $\Delta_{\min,i}$ is the minimum reward gap among the bases that include the arm $i$, $\delta_i$ is the cost gap of an arm, $r^*$ is the optimal reward, and $\omega$ is defined in Assumption 2.*

Note that the first term in (2) is related to regret from arm pulls due to selecting a suboptimal base in the round; the second term is related to arm pulls that are used to learn whether the true mean cost of an arm is greater than or less than $c$, and also the regret resulting from under-utilizing the budget; the third term is due to expected number of times the anytime constraint may be violated; the fourth term is due to suboptimal arm pulls that occur if the confidence bounds do not hold; and the last term is regret from the initialization phase. The proof of Theorem 3.1 is given in §D, and we also provide a brief proof sketch in §3.5.

Note that this problem-dependent upper bound order-wise matches the $O(K \log T)$ problem-dependent bound of prior work for the regular BwK setting. However, SUAK is not optimal for a problem-independent bound since regret can be large when the value of $\omega$ is small, i.e. if $\omega \leq 1/\log(T)$ assuming the time horizon $T$ is known. For this case, prior work such as Bernasconi et al. (2024b) can be used to achieve $O(\sqrt{KT})$ problem-independent regret in our setting.

## 3.4 RELATED WORKS

In this section, we provide some of the works that are related to our problem setting. We provide additional related works in Appendix §B.

**Bandits with Knapsacks:** The BwK problem has been studied before and algorithms that achieve optimal problem-independent regret bounds on the order of $O(\sqrt{K \cdot OPT})$ have already been developed (Badanidiyuru et al., 2013; Agrawal & Devanur, 2014). However, deriving a problem-dependent lower bound and developing algorithms that achieve this bound are still open questions. One prior work in this regard is by Sankararaman & Slivkins (2021), in which the BwK problem is studied under only one constraint; a one-dimensional cost, and no constraint on time. In this simple setting, there is a single *unique* optimal arm. They propose an algorithm that achieves a regret bound of $O(KG_{LAG}^{-1} \log T)$, where $G_{LAG}$ is defined as the Lagrangian gap of an arm. Compared to this work, our setting is more complex as we have two constraints.

Another notable work in this field is by Flajolet & Jaillet (2015), in which the BwK problem is considered under three different cases of 1, 2; and $d$ constraints. For 2 constraints, which represents a constraint on the total number of rounds and a constraint on total budget where costs of arms are one-dimensional, a regret bound of $O\left(\lambda^2 K^2 \log T/(\delta_{\min}^3 \Delta) + K^2 \sigma \log T/\delta_{\min}^3\right)$ is achieved with additional problem-dependent constants where $K$ is the number of arms; $\delta_{\min}$ is the minimum distance between the mean cost of arms and the average budget $b$; $\sigma$ is the minimum $1/\mu$ value; $\lambda = 1 + 2\kappa$; and $\kappa$ is a constant assumed to be known *a priori* such that $|\mu_i - \mu_j| \leq \kappa|\rho_i - \rho_j|$ for any $i, j$. For the $d$ constraint setting, a regret bound of $O(2^{K+d} \log T)$ is achieved. The 2-constraint setting is similar to our work, as we also have one-dimensional costs, and our anytime cost constraint can be viewed as a total budget constraint that needs to be satisfied in all rounds. It can be seen that the $O(K^2 \log T)$ regret bound in this work is not optimal for its dependence on $K^2$. In our work, we reduce this dependence on $K^2$ to $K$ while considering the more complex BwAK problem. However, our work has an increased dependence on the gap with $1/\Delta^2$ compared to the $1/\Delta$ dependence here.

Another notable prior work is by Li et al. (2021), where a $d$-dimensional cost vector is considered, with one of the dimensions of the cost vector being time. The optimal solution in this $d$-dimensional setting can be a base consisting of at most $d$ different arms. They propose a two-phase algorithm where the first phase of the algorithm pulls each arm the same number of times until the suboptimal arms are eliminated. In the second phase, the base with highest upper confidence bound is chosen. This two-phase approach greatly simplifies the theoretical analysis as the number of pulls of each individual arms is the same in the first phase. With this approach, they achieve a regret bound of $O(Kd \log T/(b^3\Delta^2) + d^4/(b^2 \min\{\chi^2, \Delta^2\} \min\{1, \sigma^2\}))$, where $\Delta$ in their setting is defined as the gap between the reward of the optimal solution per round and the maximum reward that can be obtained per round when one arm (except the null arm) is removed; $b$ is the average cost budget per round; $\chi$ is the minimum nonzero value in the optimal policy; $\sigma$ is a problem dependent constant related to the linear dependency between arms across different constraints. In our work while we have similar dependence on $\Delta$ and $K$, we have $1/\omega\delta^2$ additional dependence on cost gaps of arms. However, our setting (BwAK) is more complicated and we conjecture that these additional terms $\omega$ and $\delta^2$ are needed to satisfy the anytime constraint. The BwK setting has also been studied under different problem settings, such as in the adversarial setting (Immorlica et al., 2022), in contextual bandits (Agrawal & Devanur, 2016), under nonstationary distributions (Liu et al., 2022), and in combinatorial bandits (Sankararaman & Slivkins, 2018).

**Bandits with Replenishable Knapsacks:** In this setting, cost of an arm is allowed to be negative, which allows the knapsack to be replenished. One notable prior work is by Slivkins et al. (2024), where the contextual bandits with linear constraints (CBwLC), a more generalized version of the contextual bandits with knapsacks (CBwK) problem, which allows packing and covering constraints, as well as positive and negative resource consumption, is considered. Their algorithm also works when the initial budget is $B = \Omega(T)$, or $B = o(T)$, compared to the prior work which mostly restricts the initial budget to $B = \Omega(T)$. This is similar to our setting since our problem setting can be reduced to their problem setting by implementing the budget increase as subtracting $c$ from the costs of all arms (this also makes the skip arm in our setting have negative cost $c$ and function as the resource replenishing arm). However, their algorithm is suboptimal in our problem setting with a zero initial budget, as they remark in the discussion of (Slivkins et al., 2024, Theorem 3.6), their proposed algorithm LagrangeCBwLC achieves optimal $O(\sqrt{KT})$ regret when the initial budget $B > \Omega(T)$; and when $B = o(T)$ its regret is suboptimal. This is as expected since Lagrange-based algorithms generally require knowing the ratio $T/B$, which goes to infinity when $B = o(T)$. In our work, we are only interested in the case where the initial budget is zero, and we consider gap-dependent results instead of the gap-independent results considered here, and we propose an algorithm that achieves an order-optimal $O(K \log T)$ gap-dependent regret bound.

In Bernasconi et al. (2024a), a more general BwK formulation with long-term constraints is considered where the costs can be negative as well as positive. The long-term constraint is defined such that the total consumption of each resource at round $T$ should be less than zero up to small sublinear violations. This is again similar to this setting if we do not allow any violation of the constraint as our problem setting can be reduced to this setting again by subtracting $c$ from the costs of all arms. They show that regret is upper bounded by $O(\sqrt{KT}\log(KT))$ when the *EXP3-SIX* algorithm is used with the Primal-Dual algorithm based framework that they propose for their problem setting. They also remark that initial $o(T)$ rounds can be skipped to cover the potential violations and implement the long-term constraint as a hard constraint like in our setting. However, they provide an upper bound of $O(\sqrt{KT})$ constraint violations in (Bernasconi et al., 2024a, Corollary 8.2), which suggests that the initial $O(\sqrt{KT})$ rounds would need to be skipped to achieve hard constraints, which would lead to $O(\sqrt{KT})$ gap-independent regret in our problem setting. In our work, we show that we achieve $O(K \log T)$ gap-dependent regret for the same problem setting. In Bernasconi et al. (2024c), they consider the same problem setting as in their prior work Bernasconi et al. (2024a). Instead of a Prior-Dual algorithm based approach, they use a UCB-based approach to optimistically estimate the constraints through a weighted empirical mean of past samples. This approach lets them provide $O(\sqrt{T})$ regret in stochastic settings without assuming Slater's condition. The upper bound on constraint violations is still $O(\sqrt{KT})$, which would again lead to a $O(\sqrt{KT})$ gap-independent regret in our problem setting.

In Bernasconi et al. (2024b), there exists an arm with a negative expected cost that allows to replenish the budget. This is very similar to our setting as our case can be considered a special case of this setting that starts with zero budget. However, their work cannot be used in our setting as they assume $B = \Omega(T)$ such that $B = T\rho$, and they use the parameter $\rho$ in the Lagrangian function of the Primal-Dual algorithm template that they provide. Further, they only consider instance-independent bounds of $O(\sqrt{KT})$, and do not consider the $O(K \log T)$ instance-dependent bounds that we consider here.

**BwK with non-monotonic resource utilization:** It is a generalization of the BwK problem where in each round, a vector of resource drifts that can be positive, negative, or zero is observed along with the reward; and the budget of each resource is incremented by this drift amount (Kumar & Kleinberg, 2022). In Kumar & Kleinberg (2022), a three phase algorithm that combines the ideas in Flajolet & Jaillet (2015) and Li et al. (2021) is provided. The algorithm uses the phase one of Li et al. (2021) to identify the optimal arms, then in phase two arms are pulled to shrink the confidence intervals further, and in the third phase, the optimal arms are exploited by sampling from a perturbed distribution to ensure that the budget of each resource stays close to a decreasing sequence of thresholds. While the idea of decreasing sequence of thresholds can be seen as similar to under-budgeting in our algorithm, their problem setting assumes time horizon $T$ to be known, and threshold decays to zero over time as uncertainty decreases; however, in our setting we do not assume knowing $T$, and we incur regret from under-budgeting as we always under-budget. Their algorithm achieves $O(Km^2 \log T/(\Delta^2 \cdot \min\{\delta_{\text{drift}}^2, \sigma_{\min}^2\}))$, where $K$ is the number of arms, $m$ is the dimension of the cost vector, $\delta_{\text{drift}} > 0$ is the smallest magnitude of the drifts, and $\sigma_{\min}$ is the smallest singular value of the constraint matrix.

Comparison of our work with the prior work is summarized in Table 1. Due to different gap definitions, and different problem-dependent parameters used, we would like to note that these results are not directly comparable. Also note that a problem-dependent lower bound does not exist for the BwK problem or our BwAK problem. We remark that deriving a lower bound for BwK or BwAK would be an important future work; yet it would be challenging due to the variety of problem-dependent parameters that can be used to define the problem instance.

### 3.5 PROOF SKETCH

We now present a brief outline of the regret analysis of SUAK, which is provided in §D. In the proof, the regret is first decomposed as follows.

$$R_T \leq R_a(T) + R_b(T) + R_c(T) + R_d(T) + \sum_{t=t_{\text{init}}+1}^{T} (\mathbb{P}(\mathcal{G}^c(t)) + \mathbb{P}(\mathcal{F}^c(t))) + r^* t_{\text{init}}$$

where $R_a(T)$ is the regret from skips that are used to satisfy the anytime constraint in line 5 of Algorithm 2, and $R_b(T)$ is the regret from skips that are needed while reducing the confidence

Table 1: Comparison of our work with prior work on bandits with knapsacks

| Work | Model | Regret Bound |
|---|---|---|
| (Li et al., 2021) | Total budget | $O\left(\frac{Kd\log T}{b^3\Delta^2} + \frac{d^4}{b^2\min\{\chi^2,\Delta^2\}\min\{1,\sigma^2\}}\right)$ |
| Flajolet & Jaillet (2015) | Total budget | $O\left(\frac{\lambda^2 K^2\log T}{\delta_{\min}^3\Delta} + \frac{K^2\sigma\log T}{\delta_{\min}^3}\right)$ |
| Kumar & Kleinberg (2022) | Total budget and drift | $O\left(\frac{Km^2\log T}{\Delta^2\cdot\min\{\delta_{\mathrm{drift}}^2,\sigma_{\min}^2\}}\right)$ |
| **Our work** | Average budget | $O\left(\frac{K\log T}{\omega\delta^2\Delta^2} + \frac{K\log T}{w^2}\right) + O(1)$ |

intervals of the arm costs due to the condition in line 9 of Algorithm 2. $R_c(T)$ is due to pulls needed while reducing the confidence intervals of the arm costs (pulls from line 13 of Algorithm 2), and $R_d(T)$ is due to pulls of arms after selecting a base (pulls from line 36 of Algorithm 2); which includes the selection of suboptimal bases and regret from under-utilization of the cost budget. The terms $\sum_{t=t_{\mathrm{init}}+1}^{T}(\mathbb{P}(\mathcal{G}^c(t)) + \mathbb{P}(\mathcal{F}^c(t)))$ are due to the probability of confidence bounds not holding, and can be upper bounded as $5\pi^2 K^2/3$. The $r^* t_{\mathrm{init}}$ term is regret due to the initialization phase, we upper bound $t_{\mathrm{init}}$ by considering $K$ cost can be incurred in the initialization step in the worst case, and also noticing that the regret per round is upper bounded by $r^*$.

To upper bound $R_a(T)$, we define $t_e$ as the time the algorithm exceeds the targeted cost of $c - \log t/t$, and we define $t_f + 1$ as the time instant where the algorithm skips. Due to the design of the algorithm, the arm with the lower cost will be pulled with probability $1 - \omega$ between rounds $t_e \le t \le t_f$, and the total incurred cost between rounds $t_e \le t \le t_f$ needs to exceed the $c$ by at least $\log(t_e)$. We upper bound the probability of this event using standard concentration bounds, and apply a union bound over all possible $t_e$ and $t_f$ values to establish that $R_a(T) \le 3\pi^2 r^*/(\delta_{\min}^2)$. We upper bound $R_b(T)$ and $R_c(T)$ as follows. Using standard techniques in bandit literature, we show that an arm $i$ will be sampled at most $96\log T/\delta_i^2$ times to reduce the uncertainty in its cost estimate, and the expected regret per round will be $r^* - \mu_i$. Note that $\mu_i$ can be greater than $r^*$ for some arms, but this is balanced by skips. For arms with cost larger than $c$, we derive $104\log T/(c\delta_i)$ skips are needed.

We upper bound $R_d(T)$ as follows. For selections of a suboptimal base, using the fact that arms are sufficiently sampled by line 13 of Algorithm 2, we show that a subobtimal base $\mathcal{I} = (i,j)$ can be selected at most $\sum_{i=1}^{K} 48(\frac{\delta_i+1}{\delta_i})^2 \log T/(\Delta_{i,j}^2)$ times if it is assumed that selection of the base yields a sample of both arms in it. Due to partial observability, it will take $1/\omega$ rounds in expectation to obtain one sample for both arms. Taking this into account, we show that at most $\sum_{i=1}^{K} 48(\frac{\delta_i+1}{\delta_i})^2 \log T/(\omega\Delta_{\min,i}^2)$ pulls of arm $i$ will occur to satisfy the upper bound on the number of pulls of all bases that include arm $i$. Using the technique in (Kveton et al., 2015), we derive the worst case regret from this upper bound on the samples of arms. We also upper bound regret from cost under-utilization as $2r^*\log T/(c\omega^2)$ using $r^*/c$, the optimal reward per cost.

## 4 SIMULATIONS

We now evaluate the performance of the proposed SUAK Algorithm through simulations. For comparison, we have included the *Primal Dual* and *One Phase* Algorithms in Li et al. (2021); and the *UCB Simplex* Algorithm in Flajolet & Jaillet (2015). We would like to note that while the authors in Li et al. (2021) believe that the *One Phase* Algorithm would be optimal, they leave providing theoretical regret bounds for that algorithm as an open question claiming it would be challenging to do so. Instead, they provide theoretical guarantees for the *Primal Dual* Algorithm, which similar yet expected to have much worse empirical performance compared to the *One Phase* Algorithm. We implement the skip versions of these algorithms as described in §3.1, and we refer them by appending '_skip' after their names.

We perform simulations on the following setting with $K + 1 = 11$ arms where the mean reward and cost vectors are $\mu = [0.2, 0.25, 0.45, 0.4, 0.7, 0.75, 0.8, 0.9, 0.8, 0.7, 0]$; and $\rho = [0.2, 0.25, 0.3, 0.4, 0.6, 0.65, 0.7, 0.75, 0.8, 0.9, 0]$. Except for the null arm, the arm reward and cost values are independently sampled from a Beta distribution with parameters $\alpha = \mu * 10, \beta =$

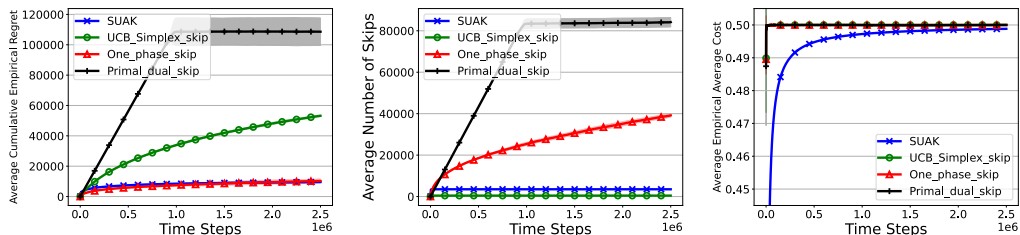

Figure 2: The plots of cumulative empirical regret (Left), number of skips (Middle), and cumulative number of skips (Right); averaged over 20 different trials.

$(1 - \mu) * 10$, where $\mu$ represents the mean of the distribution. The average cost budget per round is $c = 0.5$. We perform the simulations for 2.5 million rounds, and average over 20 different trials. The simulation results for this setting are given in Figure 2. The shaded areas in the plots represent error bars with one standard deviation. We use $\omega = 0.0625$ for SUAK.

It can be seen from the top left plot in Figure 2 that the Primal Dual Skip Algorithm performs the worst. This is as expected since the Primal Dual Algorithm is designed for theoretical performance, and pulls every arm the same number of times until finding the optimal solution. The UCB Simplex Skip also performs poorly in simulations. This is since the algorithm assumes knowledge of a constant $\kappa$ *a priori* such that $|\mu_i - \mu_j| \leq \kappa|\rho_i - \rho_j|$ for any $i, j$; and the confidence intervals for the arm rewards are multiplied by a factor of $\lambda = 1 + 2\kappa$. In the simulation setting, $\lambda = 9$; which increases the number of samples needed for exploration.

It can be seen that SUAK performs better compared to the Primal Dual or UCB Simplex algorithms, and also exceeds the performance of One Phase Skip (OPS) after around round $1.7 \times 10^6$. This is since the regret of SUAK concentrates primarily on the initial rounds, which is due to two factors. Skips needed for the under-utilization of the budget by $\log t/\omega^2$, and also pulls from the line 13 of Algorithm 2 mostly concentrate on the initial rounds. After these initial rounds, SUAK can catch up to and eventually surpass the performance of OPS due to higher regret OPS experiences from its high number of skips, verifying the practical utility of SUAK.

In terms of the number of skips, it can be seen from the bottom middle plot in Figure 2 that the number of skips is sublinear for all algorithms, and SUAK has the least number of skips. This demonstrates the effectiveness of SUAK in reducing the number of skips by under-utilizing the available budget. Note that pulls of the null arm originating from the condition in line 5 or line 9 of Algorithm 2 are counted as skips, yet pulls of the null arm when it is in the selected base is not counted as a skip. The plot on the right of Figure 2 shows the incurred average cost. As expected, SUAK starts from a smaller average cost value and approaches the per round cost budget of $0.5$ over time due to under-utilization of the budget, and other algorithms are very close to the constraint, and except the UCB Simplex Skip, need to utilize skips to avoid exceeding the constraint.

## 5 CONCLUDING REMARKS

In this paper, we introduce a previously unexplored setting for the BwK problem, which we call the bandits with anytime knapsacks (BwAK) problem; where we employ a stricter anytime cost constraint instead of a total cost budget. We provide SUAK, a novel algorithm that under-utilizes the available cost budget and uses skipping to limit the probability of violating the anytime cost constraint, and also uses upper confidence bounds to balance exploration and exploitation. SUAK achieves a regret upper bound of $O(K \log T)$ compared to the optimal solution of the linear relaxation version of the problem which does not necessarily obey the anytime cost constraint. This bound is better than the regret upper bound of prior work for the BwK setting on problem-dependent terms in a wide range of problem instances. We provide simulation results to demonstrate the empirical performance of SUAK. Our work opens multiple directions for future research. One interesting future direction is to extend our bandit results to the case where the cost of an arm is a $d$-dimensional vector. This is a challenging problem as the anytime cost constraint needs to be satisfied in every dimension, which can introduce additional skips, and hence additional regret. Another interesting open direction is the case where the distribution of rewards and costs of arms are stochastic. In this case, satisfying the anytime cost constraint would again be challenging but we conjecture it may be accomplished using a more conservative cost budget under-utilization.

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
