# OpenReview forum: "Bandits with Anytime Knapsacks"
_ICLR.cc/2025/Conference — Submitted to ICLR 2025_

### Official Review · Reviewer_BzGs · 2024-10-29

**Soundness:** 3
**Presentation:** 2
**Contribution:** 3
**Rating:** 8
**Confidence:** 3

**Summary:**

The paper studies an interesting variant of the bandits with knapsack problem, namely the bandits with anytime knapsacks. The difference is that now the constraint is imposed for any period $t$. That is, up to each period $t$, the average cost should be upper bounded by a threshold. Though the formulation is different from the bandits with knapsack problems, luckily, they admit the same linear relaxation. Thus, previous methods for bandits with knapsack problems could be modified and applied here. However, forcing the online algorithm to satisfy the anytime constraint would require additional new elements. The paper develops a novel algorithm with a carefully designed UCB principles. Further combining with a resolving LP techniques, the paper is able to show that their algorithm achieves a problem-dependent $O(\log T)$ regret. This is a new result for bandits with anytime knapsack problem and the algorithm in the paper is also new. The paper finally conducts numerical experiments to show that their algorithm performs well in practice.

**Strengths:**

1. The theoretical result is strong in that a problem-dependent $O(\log T)$ regret has been derived, in contrast to the worst-case $O(\sqrt{T})$ regret usually seen in the literature.

2. The paper conducts numerical experiments of their algorithms and delivers convincing results.

**Weaknesses:**

1. The paper only considers a setting where there is a one-dimensional cost. That is, for each period $t$, there is only one cost constraint to be satisfied. However, in the classical bandits with knapsack problems, there are usually multiple cost constraints to be satisfied.

**Questions:**

1. The paper mentions the non-degeneracy issue. However, the cost constraint considered in the paper is one-dimensional. Is non-degeneracy still an issue when the constraint is one-dimensional? Could the methods developed in the paper be further generalized to handle multi-dimensional cost constraints?

---

> ### Author Response · Authors · 2024-11-29
>
> Dear reviewer, thank you for the comments, and also for sharing this interesting reference.
>
> In the traditional LP sense, non-degeneracy issue stipulates that there exists a unique base that solves the LP. This is generally needed for multi-dimensional costs, where a mixture of different optimal bases might not be optimal, hence switching back and forth from different optimal bases might lead to suboptimal performance. Since our problem is one-dimensional, non-degeneracy in this traditional LP sense is not needed. We still make the assumption for a single optimal base for convenience in proof, this is similar to the regular multi-armed bandit problem where the existence of a single best arm is assumed. Hence, this assumption can be relaxed.
>
> However, we have a non-degeneracy issue from the cost gaps of the arms, which we define as the distance between $c$ and the mean cost of an arm. In our algorithm, we need to determine if the mean cost of an arm is less or greater than $c$ in order not to violate the anytime constraint (if an arm with a mean cost greater than $c$ is incorrectly identified as a low-cost arm it can lead to violating the constraint). For this, we sample the arms until their confidence interval does not include $c$, and necessitates us to assume $\delta_{\min} > 0$.
>
>
> Regarding your second comment, yes, the paper can be generalized to handle multi-dimensional cost constraints. The only part that needs to be generalized is the anytime constraint, since the LP part that solves for the base can already handle multi-dimensional costs. In terms of the anytime constraint, it can easily be generalized if $\omega$ is defined based on the minimum cost gap among all dimensions, though this will not be an optimal solution and have a high regret. A more optimal solution would be to define a separate $\omega$ value for each dimension, however, this would require a more involved theoretical analysis as under-budgeting applied in one dimension also affects other dimensions by altering the frequency of arm pulls within a base. Hence, while the relaxation to multiple dimensions can simply be done by defining $\omega$ based on the minimum gap among all dimensions, we leave it as a future work to implement it more  optimally.
>
> We will add the following discussion on extension to multi-dimensional costs in the final version of the paper.
>
> >  Extension to multi-dimensional costs: While we only consider one-dimensional costs in this work, this work can be extended to consider multi-dimensional costs. For such an extension, first thing that needs to be considered is generalizing the the anytime constraint to multiple dimensions. The constraint can easily be generalized if $\omega$ is the same for all dimensions, and defined based on the minimum cost gap among all dimensions. However, this will not be an optimal solution and have a high regret. A more optimal solution would be to define a separate $\omega$ value for each dimension, however, this would require a more involved theoretical analysis as under-budgeting applied in one dimension also affects other dimensions by altering the frequency of arm pulls within a base, which we leave as future work.
> >
> >
> > Another consideration is the non-degeneracy issue, where a unique optimal base is commonly assumed for problem settings with multi-dimensional costs. Relaxing this non-degeneracy assumption is considered recently in (https://arxiv.org/abs/2402.16324). They propose a novel algorithm that uses an eliminating procedure to identify the optimal base, and a resolving procedure that is adaptive to the remaining resources. A similar implementation as in this related work can be considered in our setting to prevent the need for assuming a unique optimal base in the extension to multi-dimensional costs.

---

### Official Review · Reviewer_QP71 · 2024-11-03

**Soundness:** 3
**Presentation:** 3
**Contribution:** 3
**Rating:** 8
**Confidence:** 4

**Summary:**

This work considers bandits with anytime knapsacks (BwAK), where there is an anytime cost constraint instead of a total cost budget. The authors propose a new algorithm SUAK that attains the instance-dependent regret upper bound of $O(K\ln T)$ established in prior work under the BwK framework. Numerical simulations are provided to verify the performance of SUAK.

**Strengths:**

- The paper is relatively easy to follow. Writing is mostly clear.
- The model considered is simple yet interesting and important. It has not been well studied in previous work.
- The analysis is largely sound and rigorous.

**Weaknesses:**

- Algorithm design components and novelty need further investigation.
- Assumptions might be too strong and more results might be needed to gain deeper understanding.
- Experiment details need clarification.
- Proof requires great polishing.

See Questions for details.

**Questions:**

1. SUAK Line 32: "+" should be "-"? Line 240 "The main objective of this skipping mechanism is to decouple the skips ... in practice this skipping mechanism can be ignored". Can you provide more elaboration? Since in practice it can be skipped, can you provide more formal evidence supporting the argument?
2. I am a bit confused about the algorithm design as well as the technical novelty in SUAK compared to Algorithm 3 in Kumar & Kleinberg (2022). I know the setting is a bit different, but it seems the design principle is similar especially on the "under-utilization" principle. I also want to mention another related work [1], where the authors consider an online knapsack problem with random budget replenishment (there is also an anytime budget constraint). In their setting the decision maker can only accept/reject a random cost (can be negative), and so there is no control over which "arm" to be pulled. However, the $\ln T$ buffer idea also appears (although in their setting the fluid benchmark can be too loose and different types of costs require different levels of buffer). Please provide more discussions if possible.
3. Assumption 1 and 2 implies that the decision maker has prior access on a very small $\omega \leq \delta$. It is claimed that it is necessary for meeting the anytime constraint. I think the assumption is somewhat restrictive. Can you provide more explanation why this is necessary and whether it can be relaxed?
4. The current work obtains an instance-dependent $O(\ln T)$ regret under some strong non-degeneracy assumptions as well as knowing $\omega$. What about the worst-case scenario? Is it possible to show that a policy such as OPS has a $O(\sqrt T)$ regret? Clearly OPS does not require the knowledge of $\omega$. I know generalizing to the multi-dimensional setting as in Kumar & Kleinberg (2022) can be challenging, but I think more results are needed to gain deeper understanding on the problem.
5. The paper gives $2$ simulation experiments. In the first one OPS performs much worse, while in the second one OPS performs fairly well. Why is this happening? Is it because there is some intrinsic structural difference between the two instances? Is it the case that the instance-dependent metrics $\delta$ and $\Delta$ are taking effects? Also, how do you decide $\omega$ in SUAK?
7. Technical questions. I strongly suggest the authors polish the proof. I believe the general framework is correct, but it seems there exists some confusing parts. Some examples are shown below:
 - Line 1086 - 1094: I do not think the statement is rigorous.
 - Line 1100: "≥" should be "≤"？
 - Line 1183 - 1190: I do not quite understand the proof.
 - Line 1197 - 1205: Somewhat unclear.
 - Line 1269: How do you get this from Line 1265 - 1267? Why $N_i^s(T)$ is an expected number while $N_i(T)$ is a random number?

[1] Bayesian Online Multiple Testing: A Resource Allocation Approach. arXiv preprint arXiv:2402.11425.

---

> ### Author Response · Authors · 2024-11-27
>
> Dear reviewer, thank you for the comments. We are providing our responses to each of the comments below.
>
> ### **1) Concern on the skipping mechanism**
>
>
> First, thank you, we fixed '+' to '-'. The main reason we employ skipping in SUAK is to prevent violating the anytime constraint. We check for the possibility of violating the constraint in lines 4-6 in the algorithm, and skip a round if there is any such possibility. The skipping mechanism in lines 8-10 is preventative, while there is no possibility of violating the anytime constraint we skip in that round to prevent a future skip. However, since this preventative action and the action we take if there is the possibility of violating the constraint are the same (both are skipping), it does not really matter both in practice and in theory if we skip now or later. For theoretical analysis, we separately consider the pulls from lines 8-16 and the pulls from lines 18-37 and consider their skips separately. In proof, we consider lines 4-6 to be the skipping mechanism for the lines 18-37, and we consider lines 8-10 to be a separate skipping mechanism for the lines 8-16. If the lines 8-10 did not exist, then we would need to distinguish the reason for the skip which would complicate the theoretical analysis. In simulations, while we did not report it, the algorithm performs slightly better if we ignore the skipping mechanism in lines 8-10. This is as expected since if lines 8-16 consume more than $c$ average budget and the lines 18-37 consume less than the targeted budget, there is no need to skip. However, if the lines 18-37 consume its targeted budget, then there would be the need to skip in lines 4-6, however in the worst case, the skips needed will be upper bounded by the number of skips that would have been performed in lines 8-10.
>
>
>
> We provide the relevant code blocks from the algorithm below
> ```
> 4. If S_c(t-1) + 1 > ct then
> 5.      Skip round t
> 6.      continue
> ```
>
> ```
> 8. If S_p(t-1) + 1 > c*N_p(t-1) then
> 9.      Skip round t
> 10.      continue
> ```
> Lines 8-16: To determine if the mean cost of an arm is greater than or less than c
>
> Lines 18-37: Solves the optimization problem and pulls an arm

---

> > ### Author Response · Authors · 2024-11-27
> >
> > ### **2) Concern on novelty compared to prior work**
> >
> > While it is similar to under-budgeting, the design principle of Algorithm 3 of Kumar \& Kleinberg (2022) is "sampling from a perturbed distribution to ensure the budget of each resource stays close to a decreasing sequence of thresholds" as the authors describe in the paper. The main difference is that in our setting we incur regret from under-budgeting, and hence we optimize our under-budgeting amount to satisfy the regret upper bound. In their setting, they assume they are given a time horizon $T$, and define the threshold as $c \log(T-t)$ for a constant $c>0$. Since the time horizon is known, the threshold converges to zero in round $T$, and hence regret is not incurred from this thresholding mechanism in their setting. Because of this they do not optimize the constant $c>0$ and it can be chosen as a big number (there is only a lower bound on $c$ to satisfy the regret bounds). Another difference is that their threshold term is $c \log(T-t)$, which decreases over time, while our under-budgeting amount is $\log t/\omega^2$ which increases over time. This is since they assume they are given a time horizon $T$, however we do not assume knowing the time horizon. One other difference is that the optimization problem changes in their setting based on the threshold. In our setting, the solution of the optimization problem does not depend on under-budgeting. Because of these differences, our analysis is much different from theirs.
> >
> >
> > Regarding the other related work [1], their problem setting is as follows. In each round, first the cost of rejecting the null hypothesis, which can be positive or negative, is observed, and two choices can be made; reject or accept the null hypothesis. If the hypothesis is rejected, reward is 1 and the observed cost is incurred, otherwise, the obtained reward is zero and cost is $-\alpha$. Similar to our setting, there is an anytime cost constraint, the cumulative cost needs to be less than zero at all rounds. However, this is very different from the regular bandit formulation as the rewards of the arms are constant and known, and the cost of the arm is observed at the beginning of the round before making any decision. For the case where the distribution is discrete, they propose an algorithm that uses a buffer of $c_i \log(T-t)$ where the threshold value $c_i$ depends on the cost distribution. With this algorithm, excluding the degenerate case, they achieve $O(\log T)$ regret. While the idea of a buffer can be viewed similar to our under-budgeting, similar to Kumar \& Kleinberg (2022), their time horizon $T$ is known, and the buffer zeroes out in round $T$, hence there is no regret from using the buffer. They also assume knowing the cost distribution $c_i$, and calculate the constant from this distribution, which makes the problem much easier to solve.
> >
> >
> > Hence, while the idea may be similar in principle, the implementation of under-budgeting in SUAK is much different from these two prior works due to the difference in the problem setting that we consider, and as a result of this,  theoretical analysis is also different.

---

> > > ### Author Response · Authors · 2024-11-27
> > >
> > > ### **3) The reasoning for the assumption on $\omega$**
> > >
> > > Suppose we are given a problem instance with only 3 arms, where the mean cost of arm 1 is $c+\delta_1$, the mean cost of arm 2 is $c-\delta_2$, and the third arm is the null arm; and we know that the optimal solution to the problem is a mixture of the first 2 arms (skipping is a suboptimal arm). Even in this problem instance, we need to employ under-budgeting to satisfy the anytime constraint. This is since the realized cost of an arm in a round is random; even if we pull arm 2, its realized costs can be larger than $c$, which could eventually lead to violating the cost constraint. Suppose we do under-budgeting, and target an average cost of $c - \log(t)/(\delta_2^2 t)$ in round $t$. Suppose the anytime constraint is satisfied in round $s$, i.e. the cumulative cost up to round $s$ is $cs - \log(s)/\delta_2^2$. Note that at least $\log(s)/\delta_2^2$ rounds are needed after round $s$ for the anytime constraint to be violated (This can happen if all observed costs are '1'). Suppose the we pull only arm 2 after round $s$, using Hoeffding's inequality, it can be shown that the probability of the anytime constraint being violated at round $s+u$ is upper bounded by an expression $exp(-2u\delta_2^2)$. Here, to cancel the $\delta_2^2$, $u$ needs to have a $\delta_2^2$ term in the denominator, and hence the under-budgeting needs to depend on $\delta$. Considering all possible $u,v$ values under this expression yields $O(1)$ violations on expectation (see Appendix D.6 for the more detailed proof generalized to K arms). Generalizing to $K$ arms, targeted average should be $c-\log t/ (\delta_{\min}^2 t )$ ($\delta_{\min}$ is used instead of $\delta_2$). Due to additional operations in our algorithm, our targeted cost is $c-\log t/ (\omega^2 t )$, where $\omega \leq \delta_{\min}/(2+\delta_{\min}-c)$. Note that for any $\delta_{\min}$ or $c$, it holds that   $\delta_{\min}/(2+\delta_{\min}-c) \geq \delta_{\min}/3$, hence $\omega$ can be selected as $\omega=\delta_{\min}/3$.
> > >
> > >
> > > Regarding the possibility of relaxing the assumption, yes, we believe it can be relaxed to some extent. One possibility is that the arms in the optimal base are pulled $\Omega(T)$ times, and the arms in the suboptimal bases are pulled $o(\log{T})$ times, hence the pulls of the suboptimal arms should not be expected to have a large influence in the incurred average cost. Due to this, we believe that it should be possible to use the $\delta$ value of the arms in the optimal base for $\omega$ instead of $\delta_{\min}$. To implement this, additional theoretical analysis can be made to upper bound the number of skips from suboptimal arm pulls with this changed $\delta$ value, which we leave as future work.
> > >
> > > Additionally, there can be an adaptive approach to set the  $\omega$ value each round based on the LCB of estimated $\delta_{\min}$. To see if such an approach can be implemented requires its own regret analysis, hence we leave it as future work.

---

> ### Author Response · Authors · 2024-11-27
>
> ### **4) On gap-independent (worst-case) regret**
>
> As the reviewer 8A3V pointed out, there is prior work that can achieve $O(\sqrt{KT})$ instance-independent regret in our problem setting; we have revised our paper and added the following to our prior work section:
>
>
> > In [1], a more general BwK formulation with long-term constraints is considered where the costs can be negative as well as positive. The long-term constraint is defined such that the total consumption of each resource at round $T$ should be less than zero up to small sublinear violations. This is again similar to this setting if we do not allow any violation of the constraint as our problem setting can be reduced to this setting again by subtracting $c$ from the costs of all arms. They show that regret is upper bounded by $O(\sqrt{KT}\log(KT))$ when the {\emph EXP3-SIX} algorithm is used with the Primal-Dual algorithm based framework that they propose for their problem setting. They also remark that initial $o(T)$ rounds can be skipped to cover the potential violations and implement the long-term constraint as a hard constraint like in our setting. However, they provide an upper bound of $O(\sqrt{KT})$ constraint violations in [2, Corollary 8.2], which suggests that the initial $O(\sqrt{KT})$ rounds would need to be skipped to achieve hard constraints, which would lead to $O(\sqrt{KT})$ gap-independent regret in our problem setting. In our work, we show that we achieve $O(K \log T)$ gap-dependent regret for the same problem setting.
>
>  [1] Bernasconi, Martino, Matteo Castiglioni, and Andrea Celli. "No-Regret is not enough! Bandits with General Constraints through Adaptive Regret Minimization."
>
>
> As can be seen in [1], the number of constraint violations in their setting is upper bounded by $O(\sqrt{KT})$, which suggests that if their algorithm was equipped with skips to achieve the anytime constraint that we consider here, it would need $O(\sqrt{KT})$ skips, which would lead to an instance-independent regret upper bound of $O(\sqrt{KT})$. Hence, when $\omega$ is small, it is a better idea only use skips and not utilize under-budgeting. For our algorithm, we have added the following discussion to our paper:
>
> > SUAK is not optimal if the value of $\omega$ is small, i.e. if $\omega \leq 1/\log(T)$ assuming the time horizon $T$ is known. Under this case, prior work [1] can be used to achieve $O(\sqrt{KT})$ gap-independent regret in our setting.
>
>
> ### **5) Clarification on the simulations**
> In the first simulation, we use an instance with a high $\delta_{\min}$ value to demonstrate the design idea of our algorithm. There, $\delta_{\min}=0.25$, which leads to $\omega=0.143$. In the second simulation, $\omega=0.0625$, which leads to a higher regret for SUAK. Note that the main regret term for the One Phase algorithm is skipping, and for SUAK, it is under-budgeting. For theoretical analysis, we assume we are given an $\omega$ value. For the simulations, we used $\omega=\delta_{\min}/(2+\delta_{\min}-c)$. For the case where $\delta_{\min}$ is not known, LCB of $\delta_{\min}$ can be used in implementation to update the value of  $\omega$ every round.
>
> We updated the paper and added the $\omega$ values used in the simulations. Regarding the experiment details, with this addition we believe we have provided all the details needed to be able to replicate the experiments, as we have provided our pseudo code, and the reward and cost distributions that we use.
>
>
> ### **6) Polishing of the proof**
>
> Thank you for the comment, we have made some revisions on the proof to make it more clear. Below are responses for each of your examples.
>
> Line 1086 - 1094: Thank you for pointing this out, we had several typos here; the correct relation is
> $E[N_j(t)] \geq \omega \cdot  N_{(i,j)}(t)$ instead of $E[N_j(t)] \leq  N_{(i,j)}(t)/ \omega$, we fixed this in the revision. We also fixed $\geq$ to $\leq$ in line 1100 due to this typo. The lines after 1100 are not affected from this mistake.
>
>
> Line 1183 - 1190: We corrected the proof around this part to use an upper bound on the under-budgeting amount instead of its mean amount.
>
>
> Line 1197 - 1205: We improved the proof to make it more clear.
>
> Line 1269: We corrected the proof around this part to use the upper bound on the empirical cost instead of the mean cost. The constant term in front of the second $\log T$ term in the final regret bound increased from $194$ to $202$ as a result of this.

---

> ### Comment · Reviewer_QP71 · 2024-11-30
>
> I would like to thank the authors for providing detailed response. I will keep my score for now and reconsider adjusting my score if the authors can further improve their paper. Some additional comments below:
> 1) It is clear to me now. I think the authors should include a discussion in the paper.
> 2) I think the authors should include a discussion in the paper. While I now understand the setting and implementation may be different, I do not think it is fair to claim the problem in [1] is much easier to solve. The difficulty lies in different directions. Plus, in [1] they also consider the more challenging degenerate case which is not considered in this work.
> 3) As is also pointed out by other reviewers, the assumption is crucial and can be restrictive. I think your reasoning of knowing $\omega$ is primarily focusing on the SUAK algorithm. I know relaxing it might be difficult, but I think a thorough discussion is needed in the paper for the possibility of relaxing the assumption.
>
> 4-6. Thank you for the comments. All clear to me.

---

> > ### Author Response · Authors · 2024-12-01
> >
> > Dear reviewer, thank you for the response. Our responses for your additional comments are below:
> >
> > **1)** Yes, we will include a discussion about this in the final version of the paper
> >
> > **2)** Yes, we will include the discussion in the final version of the paper. We have already revised the discussion of Kumar \& Kleinberg (2022) in the revised version of our paper as follows:
> >
> > > BwK with non-monotonic resource utilization: It is a generalization of the BwK problem where in each round, a vector of resource drifts that can be positive, negative, or zero is observed along with the reward; and the budget of each resource is incremented by this drift amount (Kumar \& Kleinberg, 2022). In Kumar \& Kleinberg (2022), a three phase algorithm that combines the ideas in Flajolet \& Jaillet (2015) and Li et al. (2021)  is provided. The algorithm uses the phase one of Li et al. (2021) to identify the optimal arms, then in phase two arms are pulled to shrink the confidence intervals further, and in the third phase, the optimal arms are exploited by sampling from a perturbed distribution to ensure that the budget of each resource stays close to a decreasing sequence of thresholds. While the idea of decreasing sequence of thresholds can be seen as similar to under-budgeting in our algorithm, their problem setting assumes time horizon $T$ to be known, and threshold decays to zero over time as uncertainty decreases; however, in our setting we do not assume knowing $T$, and we incur regret from under-budgeting as we always under-budget. Their algorithm achieves $O(Km^2 \log T / (\Delta^2 \cdot \min\{\delta_{\text{drift}}^2, \sigma_{\text{min}}^2 \}))$, where $K$ is the number of arms, $m$ is the dimension of the cost vector, $\delta_{\text{drift}} > 0$ is the smallest magnitude of the drifts, and $\sigma_{\text{min}}$ is the smallest singular value of the constraint matrix.
> >
> >
> > Regarding [1], we agree that the problem in [1] has its own difficulties, and is not much easier to solve. In our comment we meant to say that assuming to know the distribution makes it easier to come up with the constant term for the buffer. We will include the following discussion in the final version of the paper:
> >
> >
> > > In [1], a sequential hypothesis testing task is considered. In each round, first the cost of rejecting the null hypothesis, which can be positive or negative, is observed, and two choices can be made; reject or accept the null hypothesis. If the hypothesis is rejected, reward is $1$ and the observed cost is incurred, otherwise, the obtained reward is zero and cost is $-\alpha$. Similar to our setting, there is an anytime cost constraint, the cumulative cost needs to be less than zero at all rounds. This problem setting can be viewed as a bandit problem with only two arms, where one arm is rejecting the null hypothesis, and the other is accepting it. When the distribution of the cost of rejecting the null hypothesis is continuous, they propose an algorithm that requires knowing the distribution of the cost, and achieve $O(\sqrt{T})$ regret. For the case where the distribution is discrete, they propose an algorithm that uses a logarithmic buffer and threshold values to prevent the premature depletion of the resource. The threshold value depends on the cost distribution. This buffer is similar to the decreasing sequence of thresholds in (Kumar \& Kleinberg, 2022) as both use the $\log(T-t)$ term. With this algorithm, they achieve $O(\log T)$ regret for the non-degenerate case. For the degenerate case where one of the cost gaps is zero, which is more complicated to  analyze, they can achieve $O(\log^2 T)$ regret. Again, while the idea of the buffer can be seen as similar to under-budgeting in our algorithm, their problem setting assumes time horizon $T$ to be known, and the buffer decays to zero over time as uncertainty decreases; however, in our setting we do not assume knowing $T$, and we incur regret from under-budgeting as our under-budgeting term does not decay to zero.
> >
> >
> >
> > **3)** Sorry, we misunderstood this comment a bit and responded with why the $\delta$ term (through $\omega$) is needed in the regret bound. Regarding your comment on relaxing the assumption of knowing $\omega$, yes, this assumption can be relaxed. In the general response, we have provided a modified version of our algorithm that can relax this assumption. We will include this algorithm along with the simulation results and theoretical analysis on the final version of the paper.

---

> > > ### Author Response · Authors · 2024-12-02
> > > **Response requested from reviewer**
> > >
> > > Dear reviewer,
> > >
> > > As there is only less than a day left to the deadline for reviewer comments we the authors wanted to request a response from reviewer.
> > >
> > > We believe that we made a very convincing case for the strengths of our work and responded to the reviewers concerns thoroughly. We have also made the necessary improvement for the reviewer to increase their score, which was relaxing the assumption on knowing $\omega$. For this end, we have proposed an improved algorithm (posted on our general response) that relaxes the assumption on knowing $\omega$ with only minor changes compared to our previous algorithm, and provided a brief theoretical analysis on why this implementation will work. We also explained that the regret bound of this improved algorithm will be very similar to the current regret bound.
> > >
> > >
> > >
> > > In light of this, we believe we have fully addressed the concerns of the reviewer. We would like the reviewer to consider a score increase from their initial assessment if the reviewer also finds that our response adequately addresses their concerns.

---

> > > > ### Comment · Reviewer_QP71 · 2024-12-02
> > > >
> > > > I would like to thank the authors for another round of response that have addressed my concerns. I have raised my score and now advocate for accept, for the discussion on contribution and in particular for the new nice addition of addressing the unknown $\delta_{\min}$ case. I checked the roadmap and I think the idea is clear and correct. One remaining question is how the authors plan to integrate all the new materials into the future version (which might lead to a somewhat big change to the current manuscript). While I have confidence, I will leave the final decision to the chair.

---

> > > > > ### Author Response · Authors · 2024-12-03
> > > > >
> > > > > Dear reviewer,
> > > > >
> > > > > Thank you for the comment and the increase in score. Regarding the final version of the paper, we will implement the new materials with as minimal changes as possible. We plan to put this new algorithm in place of SUAK in the paper, and provide SUAK in the appendix for the special case where $\delta_{\min}$ is assumed to be known. Since in our new algorithm we keep most of the current algorithm intact; and only add two new stages, and the estimation of $\omega$; most content in the paper will stay the same with some minor additions. Below is an overview of the changes that will be made for the final version of the paper:
> > > > >
> > > > > \
> > > > > **Abstract:** no change \
> > > > > **1.** Introduction: no change \
> > > > > **2.**  Problem Statement: no change \
> > > > > **Section 3** \
> > > > > **3.1.** Naive Approach: Simulation will be rerun and results will be replaced with new algorithm \
> > > > > **3.2.** The SUAK Algorithm: Algorithm 2 will be changed (5-10 lines will be added to implement the changes), explanation on how the algorithm works will be updated accordingly (1 paragraph will be added to describe the initialization phase, 1 small paragraph will be added to describe how $\omega(t)$ is estimated). Also some discussions requested by the reviewers will be added. \
> > > > > **3.3.** Analysis of SUAK: Assumption 2, and its discussion will be deleted,
> > > > > Theorem 3.1 will be updated ($\omega$ will be replaced with $2\delta_{\min}/9$ ) \
> > > > > **3.4.** Related Works: Add  A few relevant works might be moved to appendix to gain more space \
> > > > > **3.5.** Proof sketch: Proof sketch will be added for the confidence bounds of the estimated $\omega(t)$ \
> > > > > **4.** Simulations: Simulation will be rerun and results will be replaced with new algorithm. Slight change in description of simulation setting (the part describing the $\omega$ value we used will be removed), and in discussion (discussion that the regret of SUAK concentrates in the initial rounds will be changed to reflect that the new algorithm has a separate initialization stage, discussion might also change depending on the new simulation results) \
> > > > > **5.** Concluding remarks: no change
> > > > >
> > > > > Overall, we expect these changes to increase the paper length by around half a page, which will be compensated by moving some relevant work to appendix.
> > > > >
> > > > > \
> > > > > **Changes in Appendix** \
> > > > > **A.** Table of notations: definition for the estimated $\omega(t)$ will be added \
> > > > > **B.** Additional related works: Some related works in the main paper will be moved here \
> > > > > **C.** Preliminaries: no change \
> > > > > **D.** Proof of Theorem 3.1: Regret decomposition will be revised since the regret of the new initialization stage and the newly added second stage will be considered separately \
> > > > > **Sections D.1-D.6:** no change expected (Section D.6 that considers the under-budgeting with known $\omega$ will not change, we will add an additional section on D.7 to extend to the case where  $\omega(t)$ is estimated) \
> > > > > \
> > > > > Sections that will be added: \
> > > > > **D.7:** Extend the under-budgeting analysis to the case where estimated $\omega(t)$ is used (1-2 pages) \
> > > > > **D.8:** Derivation of concentration bounds on the estimated $\omega(t)$ (1-2 pages) \
> > > > > **E:** The current version of the SUAK algorithm, its regret upper bound simulation results, and proof (will be presented as a special case where $\delta_{\min}$ is assumed to be known). (4-5 pages in total, the proof is expected to be 1-2 pages as it will reuse the reasoning and lemmas provided in section D)

---

### Official Review · Reviewer_8A3V · 2024-11-03

**Soundness:** 3
**Presentation:** 3
**Contribution:** 1
**Rating:** 5
**Confidence:** 4

**Summary:**

The paper studies the problem the problem of bandit with anytime knapsack, meaning that the knapsack constraint needs to be satisfied at all times and not only overall.

**Strengths:**

The paper is clearly written and easy to digest. The bandit with knapsack framework is interesting and this paper proposes a new model.

**Weaknesses:**

My main issue is understanding the connection with prior literature. I strongly believe that there are easy reductions from existing works (admittedly very recent) that can obtain the same results as your algorithm.

For example, take any algorithm that satisfies the constraints in high probability apart from O(sqrt(T)) violation (at any time!). Why can't you instantiate an instance of your problem with B-sqrt(T) initial budget and use any of these algorithms? Also, I'm not convinced that the skipping mechanism is not implicitly embedded in some of the existing works. In BwK you always have a default option with zero cost and zero reward, however, in more recent generalizations of the problem (usually called bandits with constraints [1,2,3]), the assumption is written slightly differently as the existence of a strictly feasible action.

I do not think this is a fundamental problem as these works are very recent. However, a real detailed discussion of the technical differences is necessary, in my opinion.

[1] Bernasconi, Martino, et al. "Beyond Primal-Dual Methods in Bandits with Stochastic and Adversarial Constraints."
[2] Bernasconi, Martino, Matteo Castiglioni, and Andrea Celli. "No-Regret is not enough! Bandits with General Constraints through Adaptive Regret Minimization."
[3] Slivkins, Aleksandrs, Karthik Abinav Sankararaman, and Dylan J. Foster. "Contextual bandits with packing and covering constraints: A modular lagrangian approach via regression."

**Questions:**

What are the technical reasons why the problem presented in this paper cannot be reduced to the more recent works on bandits with constraints, which, in the stochastic setting, provide small constraint violations?

---

> ### Author Response · Authors · 2024-11-23
>
> Dear reviewer, thank you for the comments and providing these existing works. We have updated our related work section in the paper with these relevant work you have provided, for your reference we have also included the changes below our response here.
>
>
> Regarding your first comment, yes, it is possible to instantiate the problem with $B-O(\sqrt{T})$ initial budget, or skip the first $O(\sqrt{T})$ initial rounds and use any of the algorithms you have cited or any BwK algorithm to solve our bandits with anytime knapsacks problem. However, doing so will result in $O(\sqrt{T})$ gap-dependent or gap-independent regret as this $O(\sqrt{T})$ part of the budget is unused. Using our algorithm, we can achieve a gap-dependent regret bound on the order of $O(\log T)$. This, which is our main result, is a big improvement over this naive approach. Note that our approach will also have a gap-independent regret bound of $O(\sqrt{T})$, and since this naive approach will also have a gap-independent bound of $O(\sqrt{T})$, our algorithm does not have an improvement in this part, and hence we do not report a gap-independent bound. We also would like to note that the main idea that lets us achieve the $O(\log T)$ gap-dependent regret is under-budgeting, skipping is an extra guard-rail that we use for the case where under-budgeting fails. If skipping is not one of the optimal arms, the number of skips in our algorithm is upper bounded by $O(1)$ ($O(1)$ skips in initialization and $3 \pi^2/\delta_{\min}^2$ expected number of skips from the probability that anytime constraint might be violated even though under-budgeting is used).
>
>
> Below is a copy of what we have added to the relevant works section in our paper (citation numbers follow the citation numbers in your comments):
>
> In [3], the contextual bandits with linear constraints (CBwLC), a more generalized version of the contextual bandits with knapsacks (CBwK) problem, which allows packing and covering constraints, as well as positive and negative resource consumption, is considered. Their algorithm also works when the initial budget is $B=\Omega(T)$, or $B=o(T)$, compared to the prior work which mostly restricts the initial budget to $B=\Omega(T)$. This is similar to our setting since our problem setting can be reduced to their problem setting by implementing the budget increase as subtracting $c$ from the costs of all arms (this also makes the skip arm in our setting have negative cost $c$ and function as the resource replenishing arm). However, their algorithm is suboptimal in our problem setting with a zero initial budget, as they remark in the discussion of [3, Theorem 3.6], their proposed algorithm LagrangeCBwLC achieves optimal $O(\sqrt{KT})$ regret when the initial budget $B>\Omega(T)$; and when $B= o(T)$ its regret is suboptimal. This is as expected as they require knowing the ratio $T/B$, which goes to infinity when $B= o(T)$. In our work, we are only interested in the case where the initial budget is zero, and we consider gap-dependent results instead of the gap-independent results considered here, and we propose an algorithm that achieves an order-optimal $O(K \log T)$ gap-dependent regret bound.
>
> In [2], a more general BwK formulation with long-term constraints is considered where the costs can be negative as well as positive. The long-term constraint is defined such that the total consumption of each resource at round $T$ should be less than zero up to small sublinear violations. This is again similar to this setting if we do not allow any violation of the constraint as our problem setting can be reduced to this setting again by subtracting $c$ from the costs of all arms. They show that regret is upper bounded by $O(\sqrt{KT}\log(KT))$ when the {\emph EXP3-SIX} algorithm is used with the Primal-Dual algorithm based framework that they propose for their problem setting. They also remark that initial $o(T)$ rounds can be skipped to cover the potential violations and implement the long-term constraint as a hard constraint like in our setting. However, they provide an upper bound of $O(\sqrt{KT})$ constraint violations in [2, Corollary 8.2], which suggests that the initial $O(\sqrt{KT})$ rounds would need to be skipped to achieve hard constraints, which would lead to $O(\sqrt{KT})$ gap-independent regret in our problem setting. In our work, we show that we achieve $O(K \log T)$ gap-dependent regret for the same problem setting.
>
>
> In [1], they consider the same problem setting as in [2]. Instead of a Prior-Dual algorithm based approach, they use a UCB-based approach to optimistically estimate the constraints through a weighted empirical mean of past samples. This approach lets them provide $O(\sqrt{T})$ regret in stochastic settings without assuming Slater's condition. The upper bound on constraint violations is still $O(\sqrt{KT})$, which would again lead to a $O(\sqrt{KT})$ gap-independent regret in our problem setting.

---

> > ### Author Response · Authors · 2024-12-01
> >
> > As we get close to the end of the discussion period we would greatly appreciate further discussion and questions from the reviewer and would like them to consider increasing their score if our responses have already addressed their concerns.

---

> > > ### Author Response · Authors · 2024-12-02
> > > **Response requested from reviewer**
> > >
> > > Dear reviewer,
> > >
> > > As there is only less than a day left to the deadline for reviewer comments we the authors wanted to request a response from reviewer.
> > >
> > > We believe that we made a very convincing case for the strengths of our work and responded to the reviewers concerns thoroughly. We have revised the paper with a comprehensive study on prior work, and included a detailed discussion of the technical differences of our work with the related work the reviewer has provided. We have also addressed that standard algorithms complemented with a skipping strategy or a starting budget of $B-O(\sqrt{T})$ would incur $O(\sqrt{T})$ instance-independent regret compared to $O(\log(T))$ instance-dependent regret that we achieve in our work. We also proposed an improved algorithm (posted on our general response) that relaxes the assumption on knowing $\omega$.
> > >
> > >
> > > In light of this, we believe we have fully addressed the concerns of the reviewer. We would like the reviewer to consider a score increase from their initial assessment if the reviewer also finds that our response adequately addresses their concerns.

---

### Official Review · Reviewer_VuGm · 2024-11-05

**Soundness:** 3
**Presentation:** 3
**Contribution:** 2
**Rating:** 5
**Confidence:** 4

**Summary:**

This paper studies a version of stochastic Bandit with knapsack when the budget constraint has to be respected at each time step. In particular, the average budget spent in $t$ iterations has to be at most $B \cdot T/t$.

The authors construct a learning algorithm, based on UCB methods, that provides instance-dependent regret bound with respect to the best fixed feasible distribution in hindsight. This learning algorithm performs empirically better than the trivial algorithm that simply runs any BwK algorithm and skips rounds every time the anytime constraint may be violated.

**Strengths:**

Bandits with knapsack is a relevant topic for ICLR. The paper is fairly well written, and the authors made an effort to address the obvious questions concerning their model.

**Weaknesses:**

Although somewhat natural, the idea of studying anytime constraints is pretty incremental with respect to previous work. I am not saying that the problem is immediately solvable by algorithms in the literature, but the algorithmic approach, i.e., “skipping rounds where the constraints may be violated + underspend a bit to minimize skips” is somewhat natural.

The authors did address the natural question regarding standard algorithms complemented with a skipping strategy (see Section 3.1.), but they only analyze empirically a single algorithm. It would have been more convincing to prove that any non-anytime knapsack algorithm would fail if equipped with the extra skips. Moreover, it is not clear whether such algorithms would still enjoy the instance-independent $\sqrt T$ regret bound.

Another downside of the paper is that the results are difficult to parse, given the abundance of instance-dependent parameters. Moreover, assumption 2 is not entirely motivated. Once we know $\omega$, then setting the underspending parameter is easy.

Finally, the comparison with the “replenishment” literature is not very satisfactory. The main claim is that the algorithms in that line of work do not contemplate the possibility of starting with $0$ budget. A natural fix there would be to wait for some initial rounds to build up some budget. Second, it is incorrect that \emph{only positive drift (replenishment of the resource) is considered} by Bernasconi et al. From my understanding they only require the existence of a void action that replenish the budget, which may be equivalent to the skipping action in this paper. Overall, a more comprehensive comparison is due.

**Questions:**

See above

---

> ### Author Response · Authors · 2024-11-24
>
> Dear reviewer, thank you for the comments. We have updated our paper with a more through prior work section to better motivate our approach. First, we would like to note that based on the comments of Reviewer 8A3V, we have added the following to our prior work section:
>
>
> > In [1], a more general BwK formulation with long-term constraints is considered where the costs can be negative as well as positive. The long-term constraint is defined such that the total consumption of each resource at round $T$ should be less than zero up to small sublinear violations. This is again similar to this setting if we do not allow any violation of the constraint as our problem setting can be reduced to this setting again by subtracting $c$ from the costs of all arms. They show that regret is upper bounded by $O(\sqrt{KT}\log(KT))$ when the {\emph EXP3-SIX} algorithm is used with the Primal-Dual algorithm based framework that they propose for their problem setting. They also remark that initial $o(T)$ rounds can be skipped to cover the potential violations and implement the long-term constraint as a hard constraint like in our setting. However, they provide an upper bound of $O(\sqrt{KT})$ constraint violations in [2, Corollary 8.2], which suggests that the initial $O(\sqrt{KT})$ rounds would need to be skipped to achieve hard constraints, which would lead to $O(\sqrt{KT})$ gap-independent regret in our problem setting. In our work, we show that we achieve $O(K \log T)$ gap-dependent regret for the same problem setting.
>
>
> > In [2], they consider the same problem setting as in [1]. Instead of a Primal-Dual algorithm based approach, they use a UCB-based approach to optimistically estimate the constraints through a weighted empirical mean of past samples. This approach lets them provide $O(\sqrt{T})$ regret in stochastic settings without assuming Slater's condition. The upper bound on constraint violations is still $O(\sqrt{KT})$, which would again lead to a $O(\sqrt{KT})$ gap-independent regret in our problem setting.
>
>
> >In [3], the contextual bandits with linear constraints (CBwLC), a more generalized version of the contextual bandits with knapsacks (CBwK) problem, which allows packing and covering constraints, as well as positive and negative resource consumption, is considered. Their algorithm also works when the initial budget is $B=\Omega(T)$, or $B=o(T)$, compared to the prior work which mostly restricts the initial budget to $B=\Omega(T)$. This is similar to our setting since our problem setting can be reduced to their problem setting by implementing the budget increase as subtracting $c$ from the costs of all arms (this also makes the skip arm in our setting have negative cost $c$ and function as the resource replenishing arm). However, their algorithm is suboptimal in our problem setting with a zero initial budget, as they remark in the discussion of [3, Theorem 3.6], their proposed algorithm LagrangeCBwLC achieves optimal $O(\sqrt{KT})$ regret when the initial budget $B>\Omega(T)$; and when $B= o(T)$ its regret is suboptimal. This is as expected as they require knowing the ratio $T/B$, which goes to infinity when $B= o(T)$. In our work, we are only interested in the case where the initial budget is zero, and we consider gap-dependent results instead of the gap-independent results considered here, and we propose an algorithm that achieves an order-optimal $O(K \log T)$ gap-dependent regret bound.
>
>  [1] Bernasconi, Martino, Matteo Castiglioni, and Andrea Celli. "No-Regret is not enough! Bandits with General Constraints through Adaptive Regret Minimization."
>
>  [2] Bernasconi, Martino, et al. "Beyond Primal-Dual Methods in Bandits with Stochastic and Adversarial Constraints."
>
>  [3] Slivkins, Aleksandrs, Karthik Abinav Sankararaman, and Dylan J. Foster. "Contextual bandits with packing and covering constraints: A modular lagrangian approach via regression."
>
>
> As can be seen in [1], the number of constraint violations in their setting is upper bounded by $O(\sqrt{KT})$, which suggests that if their algorithm was equipped with skips to achieve the anytime constraint that we consider here, it would need $O(\sqrt{KT})$ skips, which would lead to an instance-independent regret upper bound of $O(\sqrt{KT})$. Using our algorithm, we can achieve an instance-dependent regret bound on the order of $O(\log T)$. This, which is our main result, is a big improvement over this naive approach. Note that we do not report an instance-independent bound in our paper as SUAK will also have an instance-independent regret bound of $O(\sqrt{KT})$, which is not an improvement over this naive approach. Hence, while other algorithms complemented with some modifications might also work for our problem setting, and even achieve order-optimal instance-independent regret; the novelty of our work is proposing an algorithm with an order-optimal instance-dependent regret upper bound.

---

> > ### Author Response · Authors · 2024-11-24
> >
> > The technical motivation for Assumption 2 is as follows. The solution of the optimization problem with the constraints solved by the LP is a mixture of at most two arms. Suppose we are given arm 1 and arm 2 as a solution, where mean cost of arm 1 is $c+\delta_1$, and the mean cost of arm 2 is $c-\delta_2$; and we only pull these two arms under the anytime constraint. The strategy in our algorithm is to pull arm 2 if the empirical average cost is above the targeted cost, and pull arm 1 otherwise to reach the targeted average. Because the realized cost of an arm in a round is random, when the empirical average cost is over the targeted average, even if we pull arm 2, its realized costs can be larger than $c$, which could eventually violate the cost constraint. To prevent the violation of the anytime constraint with high probability, it can be shown that the targeted average cost for this simple problem needs to be $c-\log t/ (\delta_2^2 t )$ using the Hoeffding bounds (see Appendix C.6, where we prove this result for $K$ arms for our algorithm). Generalizing to $K$ arms, targeted average should be $c-\log t/ (\delta_{\min}^2 t )$. Due to additional operations in our algorithm, our targeted cost is $c-\log t/ (\omega^2 t )$, where $\omega \leq \delta_{\min}/(2+\delta_{\min}-c)$. Note that for any $\delta_{\min}$ or $c$, it holds that   $\delta_{\min}/(2+\delta_{\min}-c) \geq \delta_{\min}/3$, hence $\omega$ can be selected as $\omega=\delta_{\min}/3$. To implement $\omega$ in practice, if we are not given an $\omega$ and if $\delta_{\min}$ is unknown, $\omega$ can be determined using the LCB of $\delta_{\min}$ and could be updated every round (since the LCB term is used, it will converge to the true $\delta_{\min}$ over time).
> >
> >
> > We would also like to note we in fact empirically compare three algorithms complemented with a skipping strategy (see Section 4), in Section 3.1 we only provide part of the simulation results to motivate the problem.
> >
> >
> > Regarding your last comment, we are sorry to cause a confusion on the \emph{positive drift}, what we meant was, as they remark in [Assumption 2.3, Bernasconi et al.], that they use the same \emph{positive drift} assumption for the stochastic setting as in Kumar and Kleinberg, and do not consider the negative drift studied in Kumar and Kleinberg. We have corrected the paper as follows.
> >
> > > In Bernasconi et al, the bandits with replenishable knapsacks problem is considered. In this problem setting, there exists an arm with a negative expected cost that allows to replenish the budget. This setting is very similar to our setting as our case can be considered a special case of this setting that starts with zero budget. However, their work cannot be used in our setting as they assume $B=\Omega(T)$ such that $B=T\rho$, and they use the parameter $\rho$ in the Lagrangian function of the Primal-Dual algorithm template that they provide. Further, they only consider instance-independent bounds of $O(\sqrt{KT})$, and do not consider the $O(K\log T)$ instance-dependent bounds we consider here.

---

> > > ### Author Response · Authors · 2024-12-01
> > >
> > > As we get close to the end of the discussion period we would greatly appreciate further discussion and questions from the reviewer and would like them to consider increasing their score if our responses have already addressed their concerns.

---

> > > > ### Author Response · Authors · 2024-12-02
> > > > **Response requested from reviewer**
> > > >
> > > > Dear reviewer,
> > > >
> > > > As there is only less than a day left to the deadline for reviewer comments we the authors wanted to request a response from reviewer.
> > > >
> > > > We believe that we made a very convincing case for the strengths of our work and responded to the reviewers concerns thoroughly. We have revised the paper with a comprehensive study on prior work, and highlighted the differences of our work. We have addressed that standard algorithms complemented with a skipping strategy would incur $O(\sqrt{T})$ instance-independent regret compared to $O(\log(T))$ instance-dependent regret in our work. We have also proposed an improved algorithm (posted on our general response) that relaxes the assumption on knowing $\omega$ (we proposed an algorithm that does not need to know $\omega$).
> > > >
> > > >
> > > > In light of this, we believe we have fully addressed the concerns of the reviewer. We would like the reviewer to consider a score increase from their initial assessment if the reviewer also finds that our response adequately addresses their concerns.

---

### Author Response · Authors · 2024-12-01
**Relaxing the assumption on knowing $\omega$**

Dear reviewers, thank you for the reviews. Many of you commented on whether it is possible to relax the assumption on knowing $\omega$, we would like to address it here with a general response.

We have determined that, yes, this assumption can be relaxed. We provide below an improved algorithm that does not need to know an $\omega$ value in advance as input. This algorithm is a slightly modified version of SUAK that works in three stages, where first stage is used to shrink the confidence bounds of the costs. After this stage, the value of $\omega$ is estimated at every round using the LCB of $\delta_{\min}$ and we use the regular SUAK algorithm with this estimated value. We will include this algorithm in the final version of the paper, and provide its theoretical analysis and simulation results. Below, we provide the new algorithm, some brief remarks on how the proofs will be revised, and the new regret upper bound.



**Stage 1 - Initialization:**

Pull an arm $i$ as long as its confidence interval includes $c$, i.e. if $\rho_i^L(t) \leq c \leq \rho_i^U(t)$ (this is the same as line 12-16 in SUAK). The initialization continues until there is no such arm that satisfies this condition.

While pulling the arms, utilize skips to target an average cost of $c$.



**Stage 2 - Skipping:**

Set $\delta_{\min}^L(t)= \min_i  (| \bar{\rho}_i(t) - c| - \sqrt{1.5 \log t / N_i(t)})$ (find the LCB of the minimum cost gap).

Set $\omega(t) = \delta_{\min}^L(t) / (2+\delta_{\min}^L(t)-c)$.

Skip until the targeted average cost $c - \log t/ \omega^2(t) \cdot t$ is reached.


**Stage 3 - Main algorithm:**

Set $\omega(t) = \delta_{\min}^L(t) / (2+\delta_{\min}^L(t)-c)$ at the beginning of each round

Run the current version of SUAK (without the initialization phase)


### **Explanation of the changes:**
The main difference is the initialization stage. In this stage, we pull arms to shrink the confidence bounds of the costs. As a result of this, at the end at the end of the initialization stage, $\omega$ can be accurately estimated, and this estimated $\omega$ will be in the range $\frac{2}{3} \delta_{min} \leq \delta_{min}^L(t) \leq \delta_{min}$. The theoretical reason behind this range is that our LCB for the cost is defined as $$\varrho^L_i(t)  = \bar{\rho}_i(t) - 7 \sqrt{1.5 \log t / N_i(t)}, $$

and its UCB is defined similarly (this is the same definition as before). With this definition, we already showed in Appendix D.6 that $\delta_i /6 \geq \sqrt{1.5 \log t / N_i(t)}$. Using this, it can be shown that if $\rho_i > c $, $$ \left(\bar{\rho}_i(t) -  \sqrt{1.5 \log t / N_i(t)} \right) - c \geq \rho_i - c - 2 \sqrt{1.5 \log t / N_i(t)} \geq \frac{2}{3} \delta_i $$

 The case for $\rho_i < c$ is similar. Hence, $ \frac{2}{3} \delta_{min} \leq \delta_{min}^L(t) \leq \delta_{min}$.


Since we do not know the value of $\omega$ in the first stage, we target an average cost of $c$, and reduce the targeted cost to $c - \log t/ \omega^2(t) \cdot t$ in the second stage. After this stage, we implement the regular SUAK algorithm with only one change, we set $\omega(t) = \delta_{\min}^L(t) / (2+\delta_{\min}^L(t)-c)$ at the beginning of each round.




**Theoretical analysis:** In Appendix D.6, we already show that $\delta_i /6 \geq \sqrt{1.5 \log t / N_i(t)}$ due to shrinking the confidence interval. Using this result we can easily prove $\frac{2}{3} \delta_{min} \leq \delta_{min}^L(t) \leq \delta_{min}$. After this step, the only additional thing needed in the proof is to revise the under-budgeting part and show that the new $\omega(t)$ value, which is updated every round, functions correctly. Proving this is also simple since the new $\delta_{min}^L(t)$ value used in $\omega(t)$ is strictly less than or equal to $\delta_{min}$, and hence under-budgets more than needed. Thus, the new $\omega(t)$ will retain the guarantees of $O(1)$ upper bound on the number of skips needed for the possible violations of the anytime constraint while under-budgeting in the current version of the proof.



**Regret Upper Bound:** With this new implementation, $\omega$ in the current upper bound will be replaced with $\frac{2\delta_{\min}}{3(2 + 2\delta_{\min}/3 - c) }$ (it can also be replaced with $\frac{2}{9} \delta_{\min}$ since $\frac{2}{3} \frac{\delta_{\min}}{2 + 2\delta_{\min}/3 - c } \geq \frac{2}{9} \delta_{\min}$), and all other terms are expected to be the same.

---

### Meta-Review · Area_Chair_2vjd · 2024-12-22

**Metareview:**

This paper considers multi-armed bandits with knapsack constraints. The claimed difference with the existing literature is the constraint must be satisfied at each time step, rather than overall.

Two reviewers were positives, and two reviewers negative, pointing out that the contribution with respect to the current literature (safe bandits, bandit with constraints) is too small.
And indeed, I have to agree with them, the paper might be interesting and technically difficult (I mean, results are not straightforward), but based on the existing papers the reviewers mentioned, the novelty is a bit too slim for such a selective conference as ICLR.

I am sure this paper will be accepted in the future, but unfortunately, I have to recommend rejection for ICLR because of the incrementality

**Additional Comments On Reviewer Discussion:**

As mentioned above, there were positive and negative aspects on this paper, at the end it was borderline hence I went through it myself. Since I tend to agree with the spotted weaknesses, I had to decide that this paper did not reach the bar (but slightly below it)

---

### Decision · Program_Chairs · 2025-01-22

Reject